# Density Functional Theory Study of the Point Defects on KDP (100) and (101) Surfaces

**DOI:** 10.3390/molecules27249014

**Published:** 2022-12-17

**Authors:** Xiaoji Zhao, Yanlu Li, Xian Zhao

**Affiliations:** 1State Key Lab of Crystal Materials, Institute of Crystal Materials, Shandong University, Jinan 250100, China; 2Center for Optics Research and Engineering of Shandong University, Shandong University, Qingdao 266237, China

**Keywords:** KDP, defect, surface, density functional theory

## Abstract

Surface defects are usually associated with the formation of other forms of expansion defects in crystals, which have an impact on the crystals’ growth quality and optical properties. Thereby, the structure, stability, and electronic structure of the hydrogen and oxygen vacancy defects (V_H_ and V_O_) on the (100) and (101) growth surfaces of KDP crystals were studied by using density functional theory. The effects of acidic and alkaline environments on the structure and properties of surface defects were also discussed. It has been found that the considered vacancy defects have different properties on the (100) and (101) surfaces, especially those that have been reported in the bulk KDP crystals. The (100) surface has a strong tolerance for surface V_H_ and V_O_ defects, while the V_O_ defect causes a large lattice relaxation on the (101) surface and introduces a deep defect level in the band gap, which damages the optical properties of KDP crystals. In addition, the results show that the acidic environment is conducive to the repair of the V_H_ defects on the surface and can eliminate the defect states introduced by the surface V_O_ defects, which is conducive to improving the quality of the crystal surface and reducing the defect density. Our study opens up a new way to understand the structure and properties of surface defects in KDP crystals, which are different from the bulk phase, and also provides a theoretical basis for experimentally regulating the surface defects in KDP crystals through an acidic environment.

## 1. Introduction

Potassium dihydrogen phosphate (KH_2_PO_4_, KDP) crystal is the only nonlinear optical crystal material that can be used in inertial confinement nuclear fusion devices [1,2,3,4,5,6,7] because of its excellent nonlinear optical properties, high laser-damage threshold, wide light-transmission range, and ability to grow a large-sized single crystal. However, there is a certain gap between the actual nonlinear optical properties and the laser-damage threshold of KDP crystals in practical applications and for theoretical values. Numerous studies have shown that intrinsic point defects significantly influence the optical properties of KDP crystals. For example, Guillet et al. pointed out that the stress field caused by dislocation or the growth-sector boundary increases the concentration of intrinsic defects, which leads to a reduction in the laser-damage threshold [8]. The first-principles calculation results pointed out that hydrogen and oxygen defects may cause additional optical absorption and reduce the laser-damage threshold of KDP [9,10,11,12,13]. Sui et al. found that point defects such as hydrogen vacancy (V_H_), oxygen vacancy (V_O_), etc., can not only introduce defect states in the band gap but also cause large structural deformation and local lattice stress near the defect sites, which can reduce the optical performance of KDP crystals [14,15].

It should be noticed that apart from point defects in the crystals, the defects of KDP surfaces also affect the optical properties of KDP crystals. For example, by combining spectral detection and theoretical calculation, Ding et al. found that the intrinsic defects introduced by lateral cracks in the surface structure of KDP crystals have a serious effect on the laser-damage threshold of the crystals [16]. Since the surface is more active than the bulk crystal, electrons close to the defect state and surface state can be excited to high energy levels by absorbing laser energy. Moreover, when the laser-excited electrons relax to the ground state, they release energy through lattice phonons, which greatly reduces the structural stability of the KDP crystals and causes laser damage [17,18,19,20,21]. On the other hand, since the growth of KDP crystals occurs at the solid–liquid interface, and the growth environment cannot always be ideal and uniform, a large number of defects can be easily formed on the growth surfaces of KDP crystals [22,23]. Defects on the surfaces can extend into the KDP crystals as the surface layer grows further, creating large defects that may act as the origin point of the laser damage. Therefore, it is of great importance to investigate the formation and basic properties of the point defects on the growth surfaces of KDP crystals.

The KDP crystal belongs to the tetragonal crystal system with the space group of *I*$\bar{4}$ 2*d* (CCDC number: 2144930). The PO_4_ tetrahedron is the basic structural unit of the KDP crystal. These tetrahedrons are connected through hydrogen bonds and form the three-dimensional skeleton of the KDP crystal. K^+^ ions are arranged orderly in the three-dimensional skeletal system composed of PO_4_ tetrahedrons [24,25,26]. The growth surfaces of KDP crystals are (100) and (101), which correspond to the cylindrical and pyramidal surfaces, respectively. The (100) surface is a mixed layer of K ions and PO_4_ tetrahedra, while the (101) surface is a layer that exposes the K ions [27,28]. In the process of crystal growth, growth factors such as temperature, impurities, supersaturation, pH value, etc., can influence the growth quality of KDP crystals, including the formation of defects [29,30,31,32,33]. In particular, the pH value of the growth solution has been reported as an important factor affecting the growth quality and optical properties of KDP crystals [34,35,36]. For example, it has been reported that the growth quality of both the (100) and (101) surfaces of KDP crystals can be significantly improved by increasing the pH value from 3.5 to 5 [36]. Sun et al. found that increasing the pH value can increase the size and decrease the density of the scattering particles in KDP crystals, which is one of the main sources of optical loss and even crystal destruction under strong laser irradiation [37]. Although some research has been completed, the fundamental issues of point defects on KDP crystals’ surfaces are still unclear and need in-depth investigation.

In this work, we carried out density functional theory (DFT) calculations to study the local structural distortion and related electronic properties caused by V_H_ and V_O_ defects on the (100) and (101) surfaces of KDP crystals. Furthermore, by adsorbing H^+^ and OH^−^ on the surfaces, we can simulate the acid and alkali growth environments of KDP crystals and explore the influence of these different growth environments on the formation and electronic structures of the V_H_ and V_O_ defects on the crystals’ surfaces. It has been found that the vacancy defects are preferable to be formed on the surfaces rather than in the bulk KDP crystal. V_H_ and V_O_ defects are found to have different electronic structures on the (100) and (101) surfaces, which is also different from the phenomenon in the bulk KDP crystal [14,15]. In the bulk KDP crystal, both V_H_ and V_O_ defects could introduce a defect level in the band gap. On the other hand, the acidic environment is found to be conducive to the repair of V_H_ defects on KDP crystals’ surfaces, while the alkaline environment leads to more defects. Our results have produced new insights into the formation and electronic structures of the surface vacancy defects in KDP crystals. We also provide a theoretical basis for using pH value to regulate the formation and properties of point defects in KDP crystals.

## 2. Computational Details

The DFT calculations were performed by using the Vienna Ab initio Simulation Package (VASP) [38,39] with the projector-augmented-wave (PAW) formalism [40]. The H 1s^1^, O 2s^2^2p^4^, P 3s^2^3p^3^, and K 4s^1^ states were treated as valence electrons. The generalized gradient approximation (GGA) functional of Perdew Bourke Ernzerhof (PBE) [41] was used to describe the electron-exchange–correlation interactions. The electronic wave functions were expanded in plane waves using a cutoff energy of 500 eV. The force and energy convergence criteria of structural relaxation were set to be 0.02 eV/Å and 10^−5^ eV/atom, respectively. After the convergence tests, a Monkhorst-Pack k-point grid [42] of 3 × 3 × 1 was used to sample the Brillouin zone for structural optimization, and a denser grid of 7 × 7 × 1 was used for the electronic structure calculations.

According to the experimentally reported growth faces of KDP crystals [27,28], we constructed the (100) and (101) surface models by cutting four KDP molecule layers from a bulk 2 × 2 × 2 tetragonal supercell. A 15 Å vacuum layer was added perpendicular to both the (100) and (101) surfaces, to reduce the interaction between the neighboring periodic images. The surface V_H_ and V_O_ models were constructed by moving a hydrogen or oxygen atom from its original site. All the possible defect sites on the surface and sub-surface were considered, and only the defect with the lowest formation energy was chosen for the structural and property analysis. We simulated the charged defects by setting the total number of electrons and adding electrons or holes to the background of the whole system. Herein, we considered 0 and −1 charge states for the V_H_ defect and 0, +1, and +2 charge states for the V_O_ defect. The acid and alkali growth environments were simulated by placing H^+^ and OH^−^ ions at about 1.5 Å above the optimized surfaces. In order to better investigate the effect of the acid and alkali growth environments on the intrinsic point defects on the surfaces, we mainly considered the adsorption of H^+^ and OH^−^ near the defect sites of the surfaces. During the structural optimization, the bottom-most KDP molecule layer was fixed, and the top three KDP molecule layers were allowed to fully relax.

The relative stability of these defects can be determined by comparing their defect formation energies. The calculation formula is as follows [43,44,45,46]:(1)$$E_{f}\left(X^{q}\right)=E^{total}\left(X^{q}\right)-E^{total}\left(surface\right)+\underset{i}{\sum }n_{i}\mu_{i}+q\left(E_{f}+E_{v}+\Delta V\right)$$
where *E^total^* (*X^q^*) and *E^total^* (surface) are the total energies of the surface with and without the defect X, respectively. *q* represents the charge of the defect. *n_i_* represents the number of atoms of species *i* that are added or removed when the defect is created, and *μ_i_* are the corresponding chemical potentials. *E_f_* is the Fermi level of the bulk valence-band maximum *E_v_*, and Δ*V* is a correction term that aligns the reference potential in the surface with the defect, with respect to the surface without the defect. We calculated the chemical potential of the required elements by referring to the work of Sui [25], namely that the chemical potential of H and O were taken as half of the energies of H_2_ and O_2_, and the chemical potentials of P and K were calculated according to the formation enthalpies of their stable compounds P_2_O_5_ and K_2_O. The obtained chemical potential of H, O, P, and K are −3.39, −4.47, −7.39, and −3.63 eV, respectively.

## 3. Results and Discussion

### 3.1. Stability and Structures of the Point Defects on the KDP Surfaces

Firstly, we constructed the most stable structures of the (100) and (101) surfaces, which correspond to the cylindrical and pyramidal surfaces of the KDP crystals, respectively, according to the exposed surface atoms reported by previous experiments [27,28], as can be seen in Figure 1. The calculated surface energies of the (100) and (101) surfaces are 0.0321 eV/Å^2^ and 0.0327 eV/Å^2^, respectively. The slightly higher surface energy of the (101) surface indicates a faster growth rate and a higher activity than the (100) surface. This is consistent with the experimental result: the growth rate of the pyramidal surface is higher than that of the cylindrical surface [27,47]. We can also infer from the calculated surface energies that the (101) surface has a stronger adsorption capacity for ions and small molecules.

Since V_H_ and V_O_ defects have been reported as the main defects in KDP crystals and also have a significant influence on the optical properties, we then focused on the structures and electronic structures of these two point defects on the surfaces of KDP crystals. The selected sites to construct the V_H_ and V_O_ defect models are shown in Figure 1, in which they reflect the different locations and coordination environments at the surface, subsurface, and interior of the surface models. We compared the difficulty of forming V_H_ and V_O_ defects at different sites on the (100) and (101) surfaces and in the bulk KDP crystal by calculating their defect formation energies, as seen in Table 1. It can be seen that the formation energies of both the V_H_ and V_O_ defects on both the (100) and (101) surfaces are more than 1 eV lower than those in the bulk KDP. This indicates that these vacancy defects are preferable to be formed on the surfaces rather than in the bulk. Meanwhile, their formation energy on the (101) surface is about 0.3 eV lower than that on the (100) surface, which indicates that the structure and properties of the (101) surface are more sensitive and susceptible to intrinsic defects. It should be noted that the V_H_ defect prefers to form at the outermost surface site on both the (100) and (101) surfaces, while the V_O_ defect shows different stable sites on the different surfaces of KDP crystals. For the (100) surface, the V_O_ defect prefers to form at the surface site connected to the other PO_4_ group by the hydrogen bond. For the (101) surface, the formation energy of the V_O_ defect at the inner site is 1.27 eV lower than that at the surface sites, indicating that the V_O_ defects formed inside the crystal are extremely stable. In this case, the structure and properties of the (101) surface are mainly influenced by the V_H_ defect instead of the V_O_ defect. By comparing the formation energies, we selected H_surf_ and O_sub_ on the (100) surface and H_surf_ and O_in_ on the (101) surface as the V_H_ and V_O_ defect sites for the following studies.

In order to further understand the impact of surface vacancy defects on the surface structures, we compared the local lattice distortion caused by the neutral and charged V_H_ and V_O_ defects on the (100) and (101) surfaces of KDP crystals in Figure 2. The change in the related bond lengths and bond angles are listed in Table 2. For the (100) surface, the removal of a surface hydrogen atom leads to two oxygen atoms that were initially connected through this hydrogen atom (O-H…O hydrogen bond) moving away from each other, and, thus, the distance between O_1_ and O_2_ increases by 1.82 Å (Figure 2b). At the same time, the two PO_4_ tetrahedra connected by this hydrogen atom distort with the bond angle ∠O_2_P_2_O_3_ increasing by about 8°. The influence of an electron captured by the neutral V_H_ defect on the structure can be neglected (Figure 2c). For the V_O_ defect, the adjacent hydrogen atom falls into the original vacancy defect, which leads to the movement of the hydrogen atom into the adjacent hydrogen bonds toward the PO_4_ tetrahedron. The bond length of H-O_2_ is, thus, shortened by 0.12 Å (Figure 2d). The effect of a one and two electron loss from the neutral V_O_ defect on the structure can also be neglected (Figure 2e,f).

For the (101) surface, the surface V_H_ defect has little influence on the local structure of the adjacent PO_4_ groups. In contrast, the V_O_ defect in the inner layer leads to serious lattice distortion of PO_3_, because the V_O_ defect shortens the P-O_2_ bond and increases the angle of ∠O_2_PO_3_, especially for the V_O_^2+^ defect. When the neutral V_O_^0^ loses one electron to form V_O_^+^, the hydrogen atom that was originally attached to the vacancy O atom becomes attached to the adjacent PO_4_ tetrahedron, instead of occupying the vacancy site like in the case of the (100) surface, due to the electrostatic repulsive force. A similar phenomenon has also been observed in the case of V_O_^2+^. The increase in electrostatic repulsion after the loss of two electrons moves the hydrogen atom farther away from the P atom in PO_3_. As a result, the H-P distance increases by 0.43 Å compared with the structure of V_O_^+^. However, the loss of two electrons causes a serious distortion in the structure of PO_3_. For instance, the O_1_-O_2_ distance increases by 0.05 Å, the P-O_2_ distance decreases by 0.05 Å, and the bond angle ∠O_2_PO_3_ increases by ~12°, compared with the structure of V_O_^0^.

Through the above analysis, it can be concluded that both the V_H_ and V_O_ defects have relatively local influences on the structure of the (100) surface and do not cause significant surface reconstruction. However, the V_O_ defect on the (101) surface is mainly formed in the deep layers of the surface and, thus, causes a large degree and range of lattice distortion on the surface. Especially as the V_O_ charge states increase from 0 to +2, the V_O_ defect shows an increasing effect on the lattice distortion of the surface structure. We, thus, speculate that the V_O_ defect will also have a great impact on the electronic structure of the (101) surface.

### 3.2. Electronic Structures of KDP Surfaces with Vacancy Defects

To confirm our suspicions, we calculated the partial density of states (PDOS) of the (100) and (101) surfaces with different charged V_H_ and V_O_ defects, as shown in Figure 3. For both the (100) and (101) surfaces, the valence band maximum (VBM) is mainly contributed by the O 2p states, while the conduction band minimum (CBM) is mainly contributed by the K 4s states. However, the contribution of the K 4s states to the CBM of the (101) surface is much stronger than that of the (100) surface because the (101) surface is exposed by the K^+^ ions. When a V_H_^0^ is formed on the (100) surface, a defect state is introduced near the VBM, and the charge disperses in the nearby O atoms. When V_H_^0^ gains one electron, the Fermi energy level shifts slightly upward, without a significant change in the contribution from the electron states. The V_H_ on the (101) surface shows a similar influence on the VBM, but changes the contribution of the K 4s states to the CBM. Since the absence of a H atom leads to the transfer of the surrounding electrons to the defect site and a weakening of the charge transfer between the surface group and the K^+^ ions, the V_H_^0^ on the (101) surface reduces the contribution of the K electronic states to the CBM. When V_H_^0^ captures an electron to form V_H_^−^, the charge balance is restored, and the contribution of K^+^ ions at the CBM is significantly enhanced. This indicates that when the concentration of V_H_ is high enough, the electrostatic force of the K^+^ ion layer on the (101) surface is weakened. When the surface adsorbs the growth groups, reaching the equilibrium state is not favorable and, thus, affects the microscopic growth process.

The generation of V_O_ usually creates a donor level below the CBM in the bulk [13], but no donor level can be observed in the band structure of V_O_ on the (100) surface (Figure 3a). This is because after forming a V_O_ on the (100) surface, the neighboring H atom moves to the V_O_ site to form a stable H-PO_3_ configuration, which causes two additional electrons to be transferred to the H atom. When V_O_^+^ and V_O_^2+^ defects are formed, the lost electrons are distributed over the surface instead of being localized around the vacancy site. Therefore, the charged V_O_ defects do not introduce any defect levels in the band gap. In general, surface V_O_ defects can be immediately filled with H atoms, thereby reducing the effect of defects on the structure and electronic structure of the (100) surface. Therefore, the acidic environment may be more conducive to eliminate the defect state of V_O_ defects and maintain the good optical characteristics of the (100) surface.

The structure and electronic structure of the V_O_^0^ on the (101) surface are similar to those on the (100) surface due to the filling of the vacancy site by the neighboring H atom. However, when the V_O_^0^ loses one electron, the H atom leaves the V_O_ site and forms a H-O chemical bond with its nearest neighboring O atom. Therefore, a normal V_O_ defect can be observed (Figure 2k), which can introduce a deep and half-filled defect level in the band gap (Figure 3b). This defect level is composed by the 3p states of the adjacent P atom and the 2p states of the adjacent O atom. When the V_O_^+^ loses one more electron, this defect level moves below the CBM. In addition, another defect level at 2.2 eV can also be observed from the PDOS figure for the V_O_^2+^ on the (101) surface, which mainly comes from the 2p states of the three O atoms of the surface PO_3_ group. The deep defect levels introduced by V_O_^+^ and V_O_^2+^ defects can introduce additional optical absorption that damage the optical performance of KDP crystals. On the other hand, the V_O_ defect forms in the inner layer of the (101) surface, so it is difficult to repair it in the process of crystal growth, and it is easy to enter the interior of a crystal with the stability of the growth layer and the process of the crystal’s growth. Therefore, the V_O_ defect becomes the initial point of defect extension in the crystal and the prone point of laser damage, which has an adverse effect on the optical absorption and laser-damage properties of KDP crystals.

### 3.3. Effects of Acidic Environment on the Defective Surface of KDP Crystals

We simulated the effect of an acidic environment on the crystal surface by adsorbing H^+^ and compared the structural change of the perfect surfaces and defective surfaces, as shown in Figure 4. It has been found that the adsorption energies of H^+^ on the (100) and (101) surfaces are −0.43 eV and −1.08 eV, respectively. This indicates that the H^+^ is adsorbed on the (100) surface by very weak physisorption. This can be confirmed by the isolated charge distribution around the H^+^ (Figure 4a) and the fact that H^+^ is as far as 3.011 Å from the (100) surface. In order to further analyze the interaction between the adsorbed H^+^ and the surface, we plotted the PDOS of the adsorbed ions and their adjacent surface atoms in Figure 5a. Previous studies have found that if the adsorbed particles and the surface are bonded together, the PDOS of the bonded atoms will have the same resonance peaks [48,49]. As shown in Figure 5a, there is no resonance peak between H^+^ and the nearest atoms on the (100) surface, indicating that the H^+^ is not bonded to the surface. On the other hand, the much lower adsorption energy of the H^+^ on the (101) surfaces indicates that the adsorption of H^+^ on the (101) surface is quite stable. This is because H^+^ is connected to the outermost O atom on the (101) surface by a weak H-O bond. This can be judged by the charge distribution along the H-O bond (Figure 4d), by the bond length (1.674 Å) between the typical H-O bond length (0.96 Å), and by the hydrogen bond length (2.55 Å). The interaction between H and O has been proven to have a significant impact on the molecular arrangement and characteristics of materials [50]. This phenomenon can also be verified by the overlapped resonance peaks between the H ls states and O 2p states over a wide range, as shown in Figure 5a.

When the (100) surface contains a V_H_, the adsorbed H^+^ falls into the V_H_ site, and the defect is repaired. According to Figure 5a, there are resonance peaks between the H^+^ adsorbed to the surface and the O atoms on both adjacent sides, indicating that H^+^ and O atoms form chemical bonds connecting the two PO_4_ tetrahedrons. This is consistent with the significant decrease in the adsorption energy of H^+^ on the defective surface (−4.87 eV). Therefore, the PDOSs of the H^+^ adsorbed (100) surface with and without V_H_ are almost exactly the same (Figure 5b). For the surface containing the V_O_, the reduced H^+^ adsorption energy, with respect to that on the perfect surface and the overlapped resonance peaks shown in Figure 5a, all indicate that the surface V_O_ defect promotes the interaction between the adsorbed H^+^ and the surface defect. It is interesting to note that the H that fell into the V_O_ site combines with the adsorbed H^+^ and forms H_2_. Then, a Vo defect is formed at the (100) surface, and the H_2_ molecule is adsorbed at 1.23 Å above the defective surface. Therefore, a defect state of V_O_ can be clearly observed near the VBM from the PDOS in Figure 5b. It should be noted that the formation of the H_2_ molecule represents an extreme situation, because we are examining the surface under vacuum conditions. In real experimental conditions, a KDP crystal’s growth surface is exposed to the solvent environment, and the water molecules, K^+^ ions, and growth groups in the solvent weaken the interaction between the surface H atom and the adsorbed H^+^ ions, making it difficult to form a real H_2_ molecule. However, our study shows that because the surface H atom and adsorbed H^+^ ion tend to attract each other and form a stable chemical bond, the acidic environment makes the V_O_ defect on the (100) surface more stable, thereby damaging the optical properties of KDP crystals.

For the adsorption of H^+^ on the defective (101) surface, it can be seen from Figure 4e that the adsorbed H^+^ also restores the V_H_ on the (101) surface, resulting in a serious reduction in the H^+^ adsorption energy, with respect to that on the perfect surface. This phenomenon is similar to that on the (100) surface, which can be verified by the resonance peaks at the same position and with similar intensity from the PDOSs of the H^+^ adsorption on the (100) and (101) surfaces with V_H_, as shown in Figure 5a. For the H^+^ adsorption on the (101) surface with a neutral V_O_, Figure 4f shows that the H^+^ connects with the O atom in the outer layer with slight deformation, and the PO_4_ tetrahedra at the adsorption position reaches the outermost layer of the crystal instead of K^+^. The PDOS in Figure 5a shows many overlapped resonance peaks, indicating the hybridization between the H ls and O 2p states and the bonding interaction between the adsorbed H^+^ and O atoms on the surface. Although H^+^ interacts with the (101) surface, H^+^ adsorption has little effect on the electronic structure of the (101) surface with a V_O_ defect (Figure 5b). It is interesting that when the H^+^ adsorbs on the (101) surface with a charged V_O_ defect (V_O_^+^ and V_O_^2+^), the V_O_ site is filled by the adjacent H atom to form the stable H-PO_3_ configuration, as it occurs on the (100) surface. Therefore, the defect levels introduced by the charged V_O_ defects can be eliminated (Figure 5b). From the above analysis, we can see that an acidic environment can repair the surface VH defects on both the (100) and (101) surfaces and can also eliminate the defect states in the band gap of the (101) surface induced by the V_O_ defect. Therefore, the acidic environment plays a positive role in improving the surface quality and the optical properties of KDP crystals.

### 3.4. Effects of Alkaline Environment on the Defective Surface of KDP Crystals

By adsorbing OH^−^ on the (100) and (101) surfaces, we investigated the influence of an alkaline environment on the structures and electronic structures of the V_H_ and V_O_ defects on KDP crystals’ surfaces. As shown in Figure 6, the adsorption energies of OH^−^ on the (100) and (101) surfaces are −1.08 eV and −1.37 eV, respectively. From the moderate adsorption energy values for H^+^ on the (100) and (101) surfaces, we can make a preliminary inference that OH^−^ is chemically adsorbed on the (101) surface. From Figure 6a, we can see that the OH^−^ binds with the exposed H atom on the surface and forms a water molecule. The generated H_2_O molecule is adsorbed on the (100) surface and connects to the surface through a hydrogen bond. This phenomenon can be verified by the calculated PDOS in Figure 7a, where the resonance peaks of O on the OH^−^ and the surface H atom overlap at about −7.5 eV, indicating the chemical interaction between the OH^−^ and the H atom on the (100) surface. This phenomenon is consistent with the result of Zhang [51]. It can be seen from Figure 6a that a serious charge transfer occurs between the H_2_O molecule and the (100) surface, which greatly reduces the adsorption energy to −3.02 eV [52]. Therefore, the alkaline environment causes the (100) surface to lose H atoms and then destroy the surface structure to form V_H_ defects. For the adsorption of OH^−^ on the (101) surface, since there is no exposed H atom in the outermost layer, the OH^−^ adsorbs near the surface O atoms with a H-O bond length of 1.941 Å. The adsorption of OH^−^ has a slight influence on the structure and energy of the (101) surface. By comparison, it can be seen that OH^−^ has a greater impact on the (100) surface than the (101) surface. This is consistent with experiments that grew high-quality KDP crystals in a weakly acidic environment.

When OH^−^ adsorbs on the (100) surface with a V_H_ defect, OH^−^ also combines with the outmost H atom to form a H_2_O molecule and then adsorbs on the crystal surface (Figure 6b). Due to the attractive force from the V_H_, the distance between the H_2_O molecule and the surface slightly increases by 0.13 Å. Since the adsorbed H_2_O molecule has little influence on the structure and electron distribution of V_H_, the OH^−^ adsorption has little influence on the electronic structure of the (100) surface with a V_H_ defect (Figure 7b). In contrast, when OH^−^ is adsorbed on the (100) surface with a V_O_ defect, OH^−^ is far away from the surface with a distance of 3.689 Å, indicating that there is no interaction between the OH^−^ and the surface. This can also be verified by the PDOS in Figure 7a, which shows that the O atom in OH^−^ and the H atom on the surface have no overlapped resonance peaks. Therefore, the OH^−^ adsorption has little influence on the electronic structure of the (100) surface with a V_O_ defect.

For the (101) surface with a V_H_ defect, the adsorbed OH^−^ could insert into the V_H_ site, leading to an obvious charge transfer between the OH^−^ and the adjacent O atom. This can be seen from the calculated charge distribution in Figure 6e and can be verified by the many overlapped resonance peaks over a wide energy range in Figure 7a. Due to the charge transfer, the surface K^+^ contributes more electrons, and the contribution of their electronic states at the CBM is, thus, enhanced. The effect of OH^−^ adsorption on the structure and electronic structure of the (101) surface with a V_O_ defect is similar to that of the (100) surface. Both the calculated charge distribution in Figure 6f and the PDOS in Figure 7a verify that there is almost no charge transfer between OH^−^ and the surface. Therefore, the adsorption energy (−0.78 eV) is as high as that on the (100) surface. On the other hand, the introduced defect states of the charged V_O_ defects in the band gap of the (101) surface cannot be eliminated because of the very weak interaction between OH^−^ and the surface. Therefore, the alkaline environment cannot improve the electronic structure of the defective surfaces, but it leads to more V_H_ defects on the surface, which is not conducive to the improvement of the surface quality and properties.

## 4. Conclusions

In summary, we investigated the structure, stability, and electronic structure of hydrogen and oxygen vacancy defects (V_H_ and V_O_) on the (100) and (101) surfaces of KDP crystals using density functional theory. The effects of acidic and alkaline environments on the structure and properties of these defects were also studied. The results show that both V_H_ and V_O_ defects are more likely to be formed on the crystals’ surface than in the bulk phase. The effect of a V_H_ defect on the electronic structures of the (100) and (101) surfaces is almost negligible, while a V_O_ defect on the different surfaces shows different sites and electronic structure characteristics compared with that in the bulk KDP. Different from the donor energy level introduced by a V_O_ defect in the bulk KDP, a V_O_ defect on the (100) surface does not introduce any defect level and can maintain the good optical properties of the crystal. However, a V_O_ defect on the (101) surface tends to be formed in the deep layer and introduce deep defect states, resulting in extra optical absorption and a reduction in the optical properties of KDP crystals. In addition, an acidic environment is conducive to the repair of surface V_H_ and can eliminate the defect states introduced by the surface V_O_ defects, which is conducive for improving the quality of the crystals’ surface and reducing the defect density. An alkaline environment leads to the formation of more V_H_ defects, which should be avoided. Our study can provide theoretical guidance for the experimental optimization of the crystal-growth parameters needed to improve the quality of KDP crystals.

## Figures and Tables

**Figure 1 molecules-27-09014-f001:**
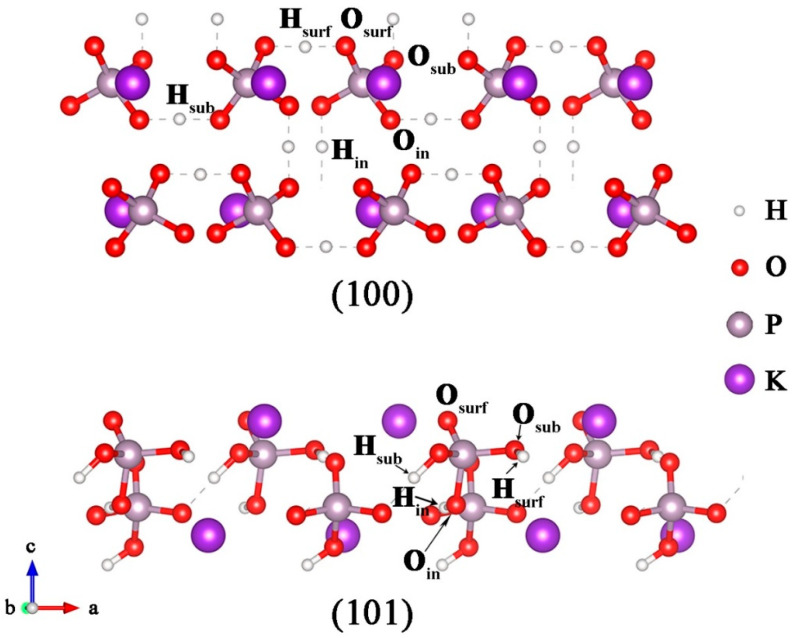
Structures of KDP (100) and (101) surfaces with two-molecule layers. The possible vacancy sites are labeled in the figures.

**Figure 2 molecules-27-09014-f002:**
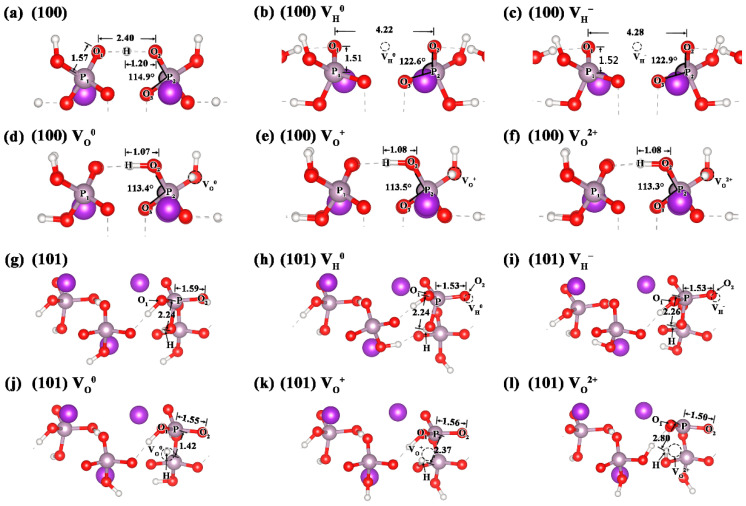
(**a**) Structure of the (100) surface of KDP crystals, local lattice distortions of neutral and charged V_H_ (**b**,**c**) and V_O_ (**d**–**f**) defects on the (100) surface of KDP crystals, (**g**) structure of the (101) surface, and local lattice distortions of neutral and charged V_H_ (**h**,**i**) and V_O_ (**j**–**l**) defects on the (101) surface of KDP crystals. The dotted circles mark the position of the vacancy defects.

**Figure 3 molecules-27-09014-f003:**
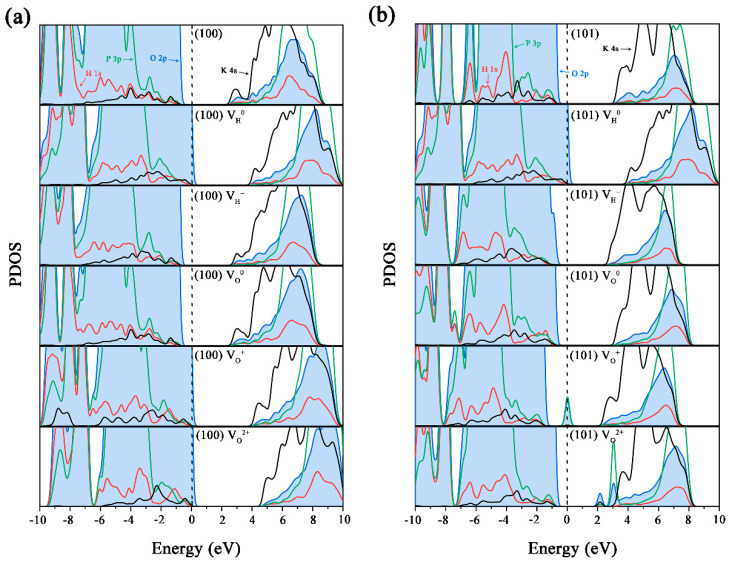
Partial density of states of the V_H_ and V_O_ defects on the (100) (**a**) and (101) (**b**) surfaces, respectively.

**Figure 4 molecules-27-09014-f004:**
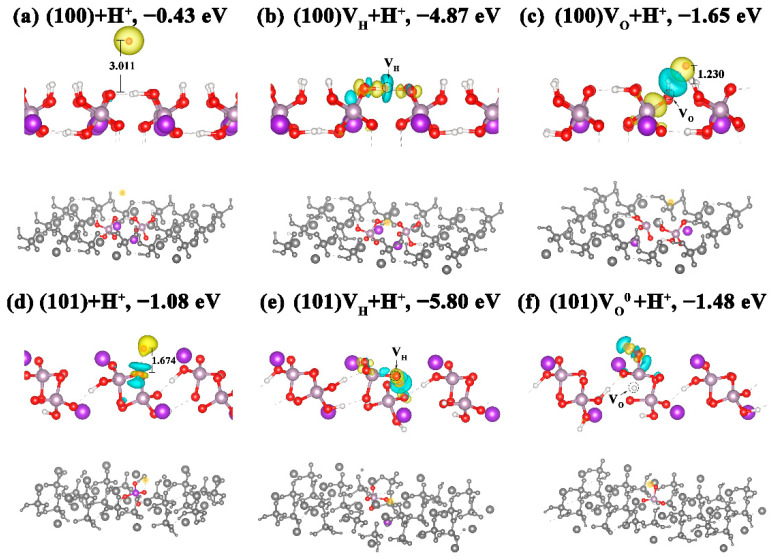
Local and overall structures as well as the charge-density difference maps of H^+^ adsorbed on (100) and (101) surfaces. (**a**) Perfect (100) surface; (**b**) (100) surface with a V_H_ defect; (**c**) (100) surface with a V_O_ defect; (**d**) perfect (101) surface; (**e**) (101) surface with a V_H_ defect; (**f**) (101) surface with a V_O_ defect. Yellow and blue regions represent electron accumulation and depletion, respectively. The corresponding adsorption energy is listed at the top of each figure.

**Figure 5 molecules-27-09014-f005:**
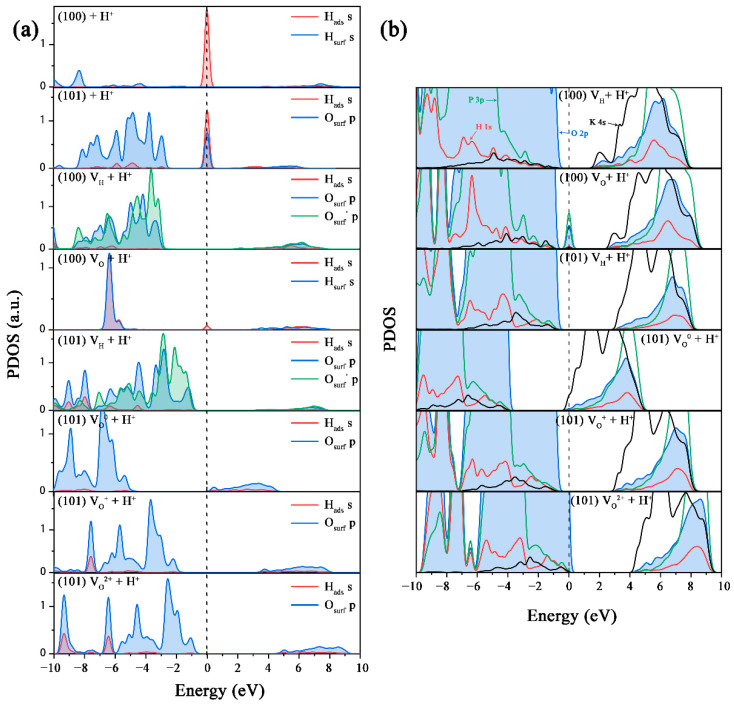
(**a**) Partial density of states of the adsorbed H^+^ ions and the adjacent surface atoms on the (100) and (101) surfaces; (**b**) partial density of states of the H^+^ adsorbed (100) and (101) surfaces with V_H_ and V_O_ defects.

**Figure 6 molecules-27-09014-f006:**
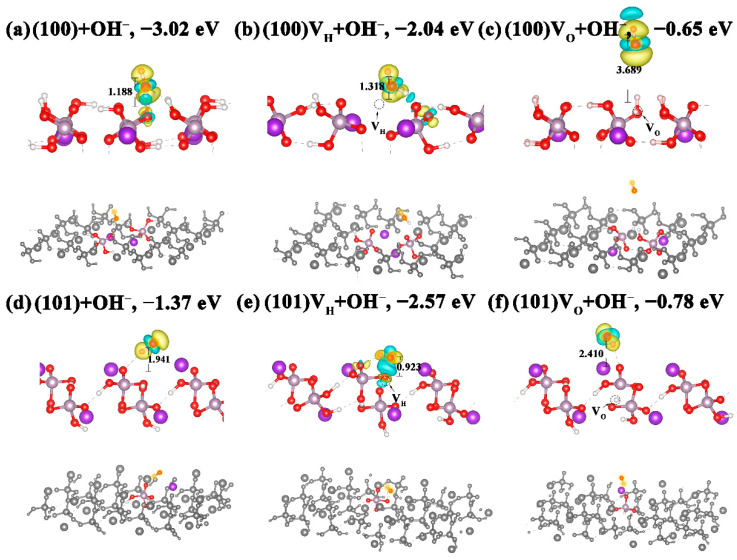
Local and overall structures as well as the charge-density difference maps of OH^−^ adsorbed on (100) and (101) surfaces. (**a**) Perfect (100) surface; (**b**) (100) surface with a V_H_ defect; (**c**) (100) surface with a V_O_ defect; (**d**) perfect (101) surface; (**e**) (101) surface with a V_H_ defect; (**f**) (101) surface with a V_O_ defect. Yellow and blue regions represent electron accumulation and depletion, respectively. The corresponding adsorption energy is listed at the top of each figure.

**Figure 7 molecules-27-09014-f007:**
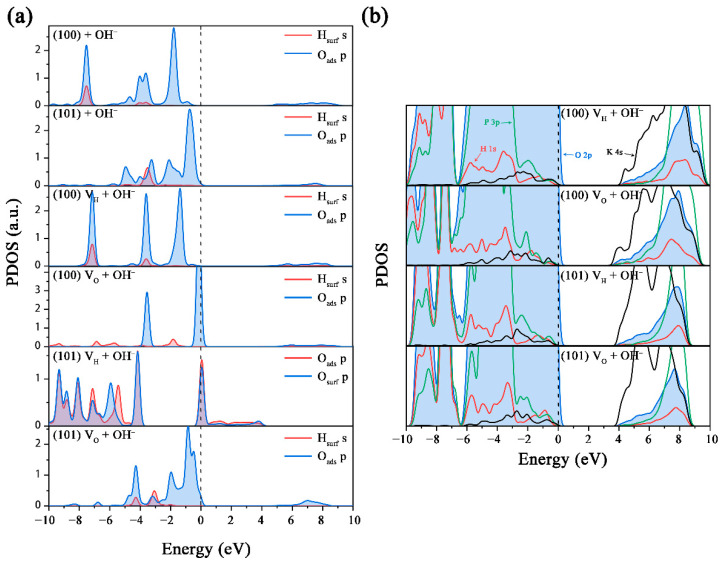
(**a**) Partial density of states of the adsorbed OH^−^ ions and the adjacent surface atoms on the (100) and (101) surfaces; (**b**) partial density of states of the OH^−^ adsorbed (100) and (101) surfaces with V_H_ and V_O_ defects.

**Table 1 molecules-27-09014-t001:** Comparison of the formation energies (eV) of V_H_ and V_O_ defects on the surfaces and in the bulk KDP crystal.

Defect	V_H_			V_O_		
Position	H_surf_	H_sub_	H_in_	O_surf_	O_sub_	O_in_
(100) surface	1.83	2.10	1.89	4.16	3.40	4.18
(101) surface	1.65	2.04	2.00	4.34	4.34	3.07
KDP bulk		2.84			4.48	

**Table 2 molecules-27-09014-t002:** Bond lengths and bond angles between atoms adjacent to the vacancy defects on the KDP crystals’ surfaces.

	Bond Length (Å)			Bond Angle
bond	H-O_2_	O_1_-O_2_	P_1_-O_1_	O_2_-P_2_-O_3_
(100)	1.20	2.40	1.57	114.9°
(100), V_H_^0^	-	4.22	1.51	122.6°
(100), V_H_^−^	-	4.28	1.52	122.9°
(100), V_O_^0^	1.07	-	-	113.4°
(100), V_O_^+^	1.08	-	-	113.5°
(100), V_O_^2+^	1.08	-	-	113.3°
bond	H-P	O_1_-O_2_	P-O_2_	O_2_-P-O_3_
(101)	2.24	2.50	1.59	104.2°
(101), V_H_^0^	2.24	2.48	1.53	105.5°
(101), V_H_^−^	2.26	2.50	1.53	105.2°
(101), V_O_^0^	1.42	2.53	1.55	106.7°
(101), V_O_^+^	2.37	2.53	1.56	106.5°
(101), V_O_^2+^	2.80	2.58	1.50	118.9°

## Data Availability

Not applicable.

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
