# Peer review of "Density Functional Theory Study of the Point Defects on KDP (100) and (101) Surfaces"

_molecules, 2022, doi:10.3390/molecules27249014_

Round 1
Reviewer 1 Report
In this work, the authors studied the structure, stability and electronic structure of hydrogen and oxygen vacancy defects (VH and VO) on the (100) and (101) growth surfaces of KDP crystals via DFT method. The results are rich and fully discussed, which is very readable for readers. In view of this, this manuscript can be accepted after minor revision.
1. Some grammatical errors in the manuscript should be brought to the attention of the author.
2. In the reference part, try to cite the high-quality literatures in the past two years.
Author Response
In this work, the authors studied the structure, stability and electronic structure of hydrogen and oxygen vacancy defects (VH and VO) on the (100) and (101) growth surfaces of KDP crystals via DFT method. The results are rich and fully discussed, which is very readable for readers. In view of this, this manuscript can be accepted after minor revision.
Q1. Some grammatical errors in the manuscript should be brought to the attention of the author.
Author reply: Thank you for the comment. As suggested by the reviewer and editor, the manuscript have been checked by a native English-speaking editing service, and the grammatical errors have been corrected.
Q2. In the reference part, try to cite the high-quality literatures in the past two years.
Author reply: As suggested by the reviewer, we added some high-quality literatures including [45] Adv. Funct. Mater. 2020, 30, 2003096, [46] ACS Energy Letters 2022, 8, 356-360 and [48] Applied Catalysis B-Environmental 2021, 298, 120531 in the revised manuscript.

Reviewer 2 Report
Decision:
Minor Revision
Comments
The authors reported a Density functional theory study of the point defects on KDP 2 (100) and (101) surfaces . this research article can be useful for the scientific community after some revision. The authors should address the following points outlined below to improve the scientific quality. After the suggested revisions are carefully addressed, this work may be considered for publication.
1. Novelty in the abstract is missing. Line 20-23 should be results with important results and finding of the paper.
2. In the introduction section author should compare the results with previous published work and highlight the novelty.
3. Line 102 author mentioned According to the experimentally reported growth faces …..Which experimental results the author is talking about? Cite relevant references here
4. Provide the CSD entry for their crystal structure.
5. Cite the following ref.
https://doi.org/10.3390/cryst11091070
6. Last there are several grammatical errors. A proofread is required.
Author Response
The authors reported a Density functional theory study of the point defects on KDP (100) and (101) surfaces. This research article can be useful for the scientific community after some revision. The authors should address the following points outlined below to improve the scientific quality. After the suggested revisions are carefully addressed, this work may be considered for publication.
Q1. Novelty in the abstract is missing. Line 20-23 should be results with important results and finding of the paper.
Author reply: The abstract has been modified according to the reviewer’s suggestion.
Q2. In the introduction section author should compare the results with previous published work and highlight the novelty.
Author reply: The introduction section has been modified according to the reviewer’s suggestion.
Q3. Line 102 author mentioned According to the experimentally reported growth faces. Which experimental results the author is talking about? Cite relevant references here
Author reply: According to the reviewer’s suggestion, we cited references [27] and [28] there.
Q4. Provide the CSD entry for their crystal structure.
Author reply: Done.
Q5. Cite the following ref.
https://doi.org/10.3390/cryst11091070
Author reply: Done.
Q6. Last there are several grammatical errors. A proofread is required.
Author reply: Thank you for the comment. As suggested by the reviewer and editor, the manuscript have been checked by a native English-speaking editing service, and the grammatical errors have been corrected.
